# DEEP QUOTIENT MANIFOLD MODELING

## ABSTRACT

One of the difficulties in modeling real-world data is their complex multi-manifold structure due to discrete features. In this paper, we propose quotient manifold modeling (QMM), a new data-modeling scheme that considers generic manifold structure independent of discrete features, thereby deriving efficiency in modeling and allowing generalization over untrained manifolds. QMM considers a deep encoder inducing an equivalence between manifolds; but we show it is sufficient to consider it only implicitly via a bias-regularizer we derive. This makes QMM easily applicable to existing models such as GANs and VAEs, and experiments show that these models not only present superior FID scores but also make good generalizations across different datasets. In particular, we demonstrate an MNIST model that synthesizes EMNIST alphabets.

## 1 INTRODUCTION

Real-world data are usually considered to involve a multi-manifold structure by having discrete features as well as continuous features; continuous features such as size or location induce a smooth manifold structure in general, whereas discrete features such as digit-class or a new object in the background induce disconnections in the structure, making it a set of disjoint manifolds instead of a single (Khayatkhoei et al., 2018). While this multiplicity makes modeling data a difficult problem, recently proposed deep generative models showed notable progresses by considering each manifold separately. Extending the conventional models by using multiple generators (Khayatkhoei et al., 2018; Ghosh et al., 2017; Hoang et al., 2018), discrete latent variables (Chen et al., 2016; Dupont, 2018; Jeong and Song, 2019), or mixture densities (Gurumurthy et al., 2017; Xiao et al., 2018; Tomczak and Welling, 2018), they exhibit improved performances in image generations and in learning high-level features.

There are, however, two additional properties little considered by these models. First, since discrete features are both common and combinatorial, there can be exponentially many manifolds that are not included in the dataset. For example, an image dataset of a cat playing around in a room would exhibit a simple manifold structure according to the locations of the cat, but there are also numerous other manifolds derivable from it via discrete variations—such as placing a new chair, displacing a toy, turning on a light or their combinations—that are not included in the dataset (see Fig. 1). Second, while the manifolds to model are numerous considering such variations, they usually have the same generic structure since the underlying continuous features remain the same; regardless of the chair, toy, or light, the manifold structures are equally due to the location of the cat.

Considering these properties, desired is a model that can handle a large number of resembling manifolds, but the aforementioned models show several inefficiencies. They need proportionally many generators or mixture components to model a large number of manifolds; each of them requires much data, only to learn the manifolds having the same generic structure. Moreover, even if they are successfully trained, new discrete changes are very easy to be made, yet they cannot generalize beyond the trained manifolds.

In this paper, we propose *quotient manifold modeling* (QMM)—a new generative modeling scheme that considers generic manifold structure independent of discrete features, thereby deriving efficiency in modeling and allowing generalization over untrained manifolds. QMM outwardly follows the multi-generator scheme (Khayatkhoei et al., 2018; Ghosh et al., 2017; Hoang et al., 2018); but it involves a new regularizer that enforces *encoder compatibility*—a condition that the inverse maps of the generators to be presented by a single deep encoder. Since deep encoders usually exhibit

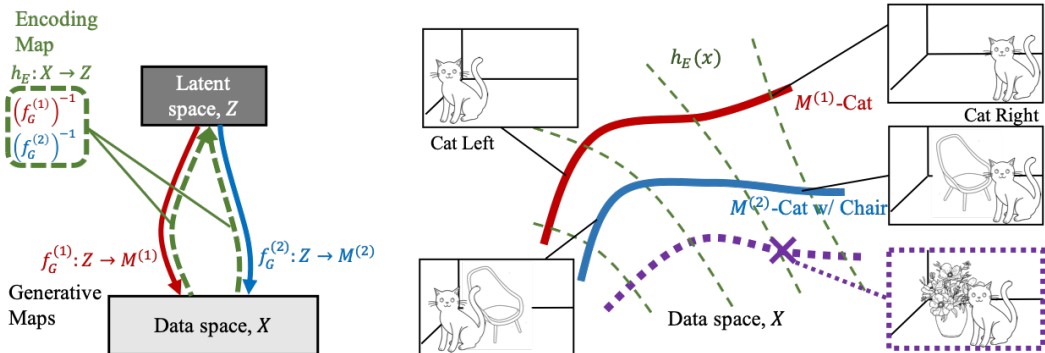

Figure 1: An illustration of quotient manifold modeling (QMM). Images of a cat moving around in a room would form an 1-D manifold due to the location of the cat; but the structure can become multi-manifold by having different discrete features (a chair or a vase in the background). The idea of QMM is to consider the generic structure shared by the manifolds using an encoding map. The encoding map induces an equivalence relation that can be seen as a contour map (green dotted curves). The quotient of the relation (or the orthogonal curve to the map) gives the manifold structure for an untrained image (shown in purple, dotted).

good generalizability, this condition not only makes a generic structure be shared among the generators but also makes it generalizable to untrained manifolds. In particular, it induces a generalizable equivalence relation between data, and the manifold structure of out-of-sample data can be derived by taking the quotient of this relation, hence the name QMM.

Since the implementation of QMM is essentially adding a regularizer, it can be easily applied to existing deep generative models such as generative adversarial networks (GANs; (Goodfellow et al., 2014)), variational auto-encoders (VAEs; (Kingma and Welling, 2013)), and their extensions. We demonstrate that these QMM-applied models not only show better FID scores but also show good generalizations.

Our contributions can be summarized as follows:

- We propose QMM, a new generative modeling scheme that considers generic manifold structure, thereby allowing generalizations over untrained manifolds.

- We derive a regularizer enforcing encoder compatibility, an essential condition for QMM.

- We show that GANs and VAEs implementing QMM show superior FID scores and generalize across different datasets.

## 2 BACKGROUND

### 2.1 MANIFOLD MODELING IN GANS AND VAES

While generative adversarial networks (GANs) (Goodfellow et al., 2014) and variational auto-encoders (VAEs) (Kingma and Welling, 2013) are two different strands of models, they have the same data-modeling scheme that leads to a manifold structure (though VAEs involve stochasticity). They model data $x \in X$ as a transformation of a low-dimensional latent code $z \in Z$ via a generative (decoding) map $f_G : Z \to X$, which makes every datum they consider lie on a subspace $M = f_G(Z) \subset X$. Since $f_G$ can be assumed to be smooth and injective in practice (Shao et al., 2017), $M$ accords with the mathematical definition of a smooth manifold.

But to deal with multi-manifold data, these models need to approximate disconnections in the structure with low densities. This requires them to have a highly nonlinear $f_G$, which is difficult to learn and often leads to either a low-quality model or a mode collapse (Khayatkhoei et al., 2018).

## 2.2 Multi-manifold Extensions

To better model the multi-manifold structure, several studies proposed extended GAN and VAE models that consider each of the manifolds separately. According to which component is extended, the approaches can be broken down into the below three. While these approaches have advantages in dealing with multi-manifold data, they still show limiting performance in learning generic structure and do not allow generalization over untrained manifolds.

*1) Multi-generators*—$x^{(i)} = f_G^{(i)}(z)$ (Khayatkhoei et al., 2018; Ghosh et al., 2017; Hoang et al., 2018). In this approach, each manifold is modeled by a separate generator. The generators are usually independent, but some models design them to share the weight parameters in a subset of the layers. This in part contributes to the learning of a generic structure, but lacks theoretical grounds and shows inferior performances (see Appendix F). *2) Mixture density*—$x^{(i)} = f_G(z^{(i)})$, where $z^{(i)} \sim p^{(i)}$ (Gurumurthy et al., 2017; Xiao et al., 2018; Tomczak and Welling, 2018). In this approach, each manifold is modeled by a separate mode of the latent distribution. While the modes outwardly share the generator, the actual mappings are effectively different from each other as they reside in different regions in $Z$. *3) Discrete latent variables*—$x^{(i)} = f_G([z; d])$ (Chen et al., 2016; Dupont, 2018; Jeong and Song, 2019). In this approach, discrete random variables are explicitly defined and concatenated to the continuous variable. Since discrete information is slowly blended in layer by layer, it can learn the generic structure to some degree, but not as clear (see Table 1).

## 3 Quotient Manifold Modeling (QMM)

QMM inherits the multi-generator scheme (Khayatkhoei et al., 2018; Ghosh et al., 2017; Hoang et al., 2018), but involves an additional regularizer enforcing the *encoder compatibility*. Leaving the regularizer for the next section, we first explain the role of this compatibility as binding the generative maps. Then, we see how a plausible equivalence relation can be defined using a deep encoder. Lastly, we explain how a new manifold can be obtained by taking the quotient of the relation.

### 3.1 Encoder Compatibility

**Definition 1.** Let $\mathcal{H}$ be a set of encoding maps $(X \to Z)$ that can be represented by a deep encoder. We say that generative maps $\{f_G^{(i)} : Z \to M^{(i)} \subset X\}_{i=1}^A$ have *encoder compatibility* if there exists $h_E \in \mathcal{H}$ satisfying $(f_G^{(i)})^{-1}(x) = h_E(x)$ for all $x \in M^{(i)}$ and $i$.

With this condition satisfied, the generative maps $\{f_G^{(i)}\}_i$ are no longer independent to each other but share a single $X \leftrightarrow Z$ translation rule represented by the deep encoder $h_E \in \mathcal{H}$. However, this binding is meaningful only when $\mathcal{H}$ has a certain property; otherwise, $h_E$ is just an *extension* of functions $\{(f_G^{(i)})^{-1}\}_i$ giving no useful signal to $\{f_G^{(i)}\}_i$.

In practice, $\mathcal{H}$ indeed involves an important property that its elements—deep encoders—have good generalizability. Having numerous parameters, deep encoders could overfit data, but in practice they find the smoothest function exhibiting generalizability. For example, if a deep encoder is trained on images illustrated in Fig. 1 to output the position of the cat, we expect it would work fairly well even after we place a vase or turn on a light, generalizing over the discrete changes of the room condition. While this generalizing property is not fully understood to date (there are several compelling theories (Zhang et al., 2016; Brutzkus et al., 2018)), it has been consistently demonstrated by many deep recognition models (e.g., VGG-Net (Simonyan and Zisserman, 2014), Fast R-CNN (Girshick, 2015)). In this regard, we assume its continuity and generalizability in developing our model, and verify them later from the experiments.

### 3.2 Equivalence Relation and Quotient Manifolds

Putting the generalizability of $h_E$ and the compatibility $(f_G^{(i)})^{-1} = h_E$ together, we expect $h_E$ to output the continuous features $z$ given data $x$. Then, there is a naturally induced equivalence relation

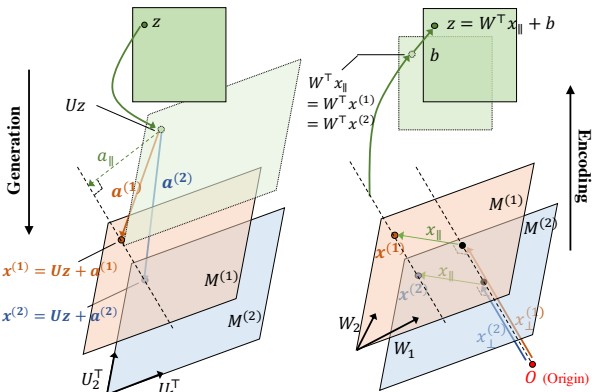

Figure 2: The generators and the encoder of a linear layer are illustrated. **Generation:** A latent point $z$ is transformed by the same weight $U$, then different biases $a^{(1)}$ and $a^{(2)}$ are added to generate $x^{(1)}$ and $x^{(2)}$ for different manifolds. The biases are regularized to have *the same tangential component* (to the column space of $W$ or $U^\top$) $a_\parallel$, which makes the generators encoder-compatible (Def. 1). **Encoding:** The points $x^{(1)}$ and $x^{(2)}$ lying on different manifolds are mapped to the same $z$ if they have the same tangential components $x_\parallel$. Note this becomes false if the biases are not regularized.

(called the *kernel* of $h_E$) $x^{(1)} \sim_{h_E} x^{(2)} \iff h_E(x^{(1)}) = h_E(x^{(2)})$, which effectively groups data having the same continuous features together regardless of the manifolds they are lying on.

This can be seen more concretely in Fig. 1. Assuming the model is well trained, it should have captured the crucial continuous feature—the location of the cat—in $Z$. Due to the encoder compatibility, such a feature should be encoded in $Z$ smoothly, meaning that $h_E$ outputs the location information of the cat in a consistent and generalizable way, if not exactly calibrated. Given this, images having the cat on the same location, regardless of the chair, will have the same $h_E(x)$, thus being equivalent under $\sim_{h_E}$ (ones on a green dotted curve). Since $h_E$ can generalize, the equivalence relation is consistent for data lying on an untrained manifold as well (ones with the vase, drawn in purple).

Now that the data lying on different manifolds are made equivalent under $\sim_{h_E}$, manifold structures are described as quotients of this relation. In implementations, taking the quotient is the same as taking the *orthogonal* directions to the equivalence-relation contours. In Fig. 1, we can see that $M^{(1)}$ and $M^{(2)}$, the manifolds that are included in the dataset, are already orthogonal to the contours formed from $h_E$. When given an untrained image (shown in purple), we can obtain the new manifold just by following the orthogonal directions. It will be explained later that this manifold can be described by a new generator, whose bias parameters are optimized for the manifold to pass through the given image (see Sec. 5).

## 4 BIAS-ALIGNING REGULARIZER

To implement the discussed scheme, the main issue is to make the generators have the encoder compatibility. We first examine a simplified case where each generator is single-layered and derive that a sufficient condition for this is that the biases of the generators are aligned in a certain way (Proposition 1). To achieve the alignment, we introduce a *bias-aligning regularizer*, the main component of QMM. After, we explain how the regularizer can be extended to the multi-layer case.

### 4.1 ENCODER COMPATIBILITY FOR SINGLE LINEAR LAYER

Consider a set of linear generative maps $\{f_G^{(i)} : Z \to M^{(i)} \subset X\}_i$; each of the maps is defined as $f_G^{(i)}(z) := U^{(i)}z + a^{(i)}$, where $U^{(i)} \in \mathbb{R}^{d_X \times d_Z}$ ($d_X > d_Z$) and $a^{(i)} \in \mathbb{R}^{d_X}$ are the weight and bias parameters respectively. We assume $U^{(i)}$ is a full-rank matrix (rank-$d_Z$) such that $f_G^{(i)}$ is injective. Then, the inverse $(f_G^{(i)})^{-1} : M^{(i)} \to Z$ can be derived as

$$(f_G^{(i)})^{-1} = (U^{(i)})^+ (x^{(i)} - a^{(i)}) \tag{1}$$

where $(U^{(i)})^+ := \left((U^{(i)})^\top U^{(i)}\right)^{-1} (U^{(i)})^\top$ denotes the pseudo-inverse of $U^{(i)}$. To achieve the encoder compatibility (Def. 1), our desire is to restrict the inverse maps $\{(f_G^{(i)})^{-1}\}_i$ such that they can be represented by a single encoder $h_E : X \to Z$. One simple way to achieve this is to use the following proposition.

**Proposition 1.** *If the linear generating maps $\{f_G^{(i)}(z)\}_i$ are restricted to have the same weight $U$ and to have the same tangential components of bias $a_\parallel$, then their inverses $\{(f_G^{(i)})^{-1}\}_i$ can be represented by a single linear encoder $h_E(x) := W^\top x + b$, where $W = U(U^\top U)^{-1}$ and $b = -W^\top a_\parallel$.*

*Proof.* Let $a_\parallel^{(i)}$ and $a_\perp^{(i)}$ denote the tangential and the normal components of $a^{(i)}$ (to the column space of $U^\top$), respectively. Then, the restrictions can be expressed as $U^{(i)} = U$ and $a^{(i)} = a_\parallel + a_\perp^{(i)}$ for all $i$. Substituting these in Eq. 1,

$$
\begin{aligned}
(f_G^{(i)})^{-1} &= U^+ x - U^+(a_\parallel + a_\perp^{(i)}) = U^+ x - U^+ a_\parallel \\
&= W^\top x + b. \qquad \square
\end{aligned}
$$

## 4.2 BIAS-ALIGNING REGULARIZER FOR SINGLE LINEAR LAYER

When implementing Proposition 1, making $\{U^{(i)}\}_i$ the same is as trivial as setting the same weight $U$ for all $f_G^{(i)}$, but making $\{a_\parallel^{(i)}\}_i$ the same is nontrivial since the tangential direction keeps changing while training. One solution would be to use a regularizer minimizing the sum of the variance: $\text{trace}(\text{cov}(a_\parallel^{(i)}))$. However, computing this term is intractable due to the inversion $(U^\top U)^{-1}$ inside of $a_\parallel^{(i)} = U(U^\top U)^{-1} U^\top a^{(i)}$.

**Theorem 1.** *The following inequality holds:*

$$
trace\left(cov(U^\top a^{(i)})\right) \geq \frac{1}{d_z} H(\{\lambda_k\}_{k=1}^{d_z}) trace\left(cov(a_\parallel^{(i)})\right)
$$

*where $\{\lambda_k\}_{k=1}^{d_z}$ denotes the eigenvalues of $U^\top U$ and $H(\cdot)$ denotes harmonic mean.*

*Proof.* See Appendix A. $\qquad \square$

As the harmonic mean in Theorem 1 is constant from the perspective of $a_\parallel^{(i)}$, we can minimize the original term by minimizing the upper bound instead. With an addition of $\log$ to match the scale due to the dimensionality of the layer, we propose this upper bound as a regularizer to make $a_\parallel^{(i)}$ the same:

$$
\textbf{BA-regularizer:} \quad R_{BA} = \log\left(\text{trace}\left(\text{cov}(U^\top a^{(i)})\right)\right). \tag{2}
$$

## 4.3 MULTI-LAYER NETWORK

The encoder compatibility for multi-layer networks can be enforced straightforwardly by applying Proposition 1 to all the linear layers. That means, for the $l$-th linear layer of all the generators, their weights are shared, $U_l^{(i)} = U_l$, and their biases are regularized to have the same tangential components, $a_l^{(i)} = a_{l,\parallel} + a_{l,\perp}^{(i)}$ via Eq. 2; other layers—nonlinear activation functions (we use LeakyReLU) and batch-normalization layers—are simply set to be shared.

This design guarantees the inverses of the generator networks to be represented by a single deep encoder (though we do not actually compute the inversion), inducing the encoder compatibility. Since all the layers are invertible, the entire networks are invertible; also, due to Eq. 2, inverses of $l$-th linear layers can be represented by a single linear layer ($W^{(l)\top} x + B$), and other layers can be trivially inverted and represented by the same layers.

Table 1: FID (smaller is better) and Disentanglement (larger is better) scores are shown. We compare WGAN (Arjovsky et al., 2017), DMWGAN (Khayatkhoei et al., 2018), $\beta$-VAE (Higgins et al., 2016), InfoGAN (Chen et al., 2016) with our model. The mean and std. values are computed from 10 (MNIST) and 5 (3D-Chair) replicated experiments.

| | | GANs | | | | | VAEs | | |
|---|---|---|---|---|---|---|---|---|---|
| | | WGAN | DMWGAN | InfoGAN | Q-WGAN (Ours) | Q-WGAN, $\lambda = 0$ | $\beta$-VAE | Q-$\beta$-VAE (Ours) | Q-$\beta$-VAE, $\lambda = 0$ |
| **FID** | MNIST | $10.13 \pm 3.16$ | $\mathbf{5.41 \pm 0.34}$ | $12.17 \pm 1.30$ | $\mathbf{5.69 \pm 0.89}$ | $15.74 \pm 10.00$ | $58.43 \pm 0.23$ | $\mathbf{41.74 \pm 1.40}$ | $40.74 \pm 2.03$ |
| | 3D-Chair | $125.32 \pm 1.16$ | $184.5 \pm 31.5$ | $187.94 \pm 9.51$ | $\mathbf{125.27 \pm 4.34}$ | $128.44 \pm 7.06$ | $217.12 \pm 0.55$ | $\mathbf{211.39 \pm 7.42}$ | $210.05 \pm 4.26$ |
| **Disentangle** | MNIST (slant) | $1.62 \pm 0.41$ | $1.08 \pm 0.04$ | $1.24 \pm 0.15$ | $\mathbf{2.15 \pm 0.17}$ | $1.76 \pm 0.35$ | $5.04 \pm 1.19$ | $\mathbf{5.93 \pm 1.78}$ | $5.18 \pm 0.77$ |
| | MNIST (width) | $1.68 \pm 0.49$ | $1.11 \pm 0.06$ | $1.18 \pm 0.11$ | $\mathbf{2.93 \pm 0.60}$ | $2.75 \pm 0.67$ | $5.63 \pm 0.75$ | $\mathbf{5.71 \pm 1.06}$ | $5.00 \pm 0.77$ |
| | 3D-Chair (height) | $2.14 \pm 0.20$ | $1.14 \pm 0.05$ | $1.41 \pm 0.34$ | $\mathbf{3.27 \pm 1.73}$ | $2.76 \pm 0.31$ | $8.10 \pm 0.20$ | $\mathbf{8.88 \pm 1.23}$ | $6.99 \pm 1.20$ |
| | 3D-Chair (bright.) | $3.53 \pm 0.80$ | $1.20 \pm 0.14$ | $3.02 \pm 0.88$ | $\mathbf{4.45 \pm 0.66}$ | $4.24 \pm 0.54$ | $3.96 \pm 0.20$ | $\mathbf{7.27 \pm 1.48}$ | $5.88 \pm 1.01$ |

## 5 DEEP QUOTIENT GENERATIVE MODELS

Now that we described the QMM scheme, we explain how it can be applied to the concrete models. We present its applications on Wasserstein GANs and $\beta$-VAEs among others.

**Generation** As above, let us denote the weight of the $l$-th linear layer as $U_l$ and the biases as $\{a_l^{(i)}\}_{i=1}^A$ . Then, we can express the data generating distribution in the form of ancestral sampling:

$$x \sim f_G^{(i)} \left( z; \{U_l, a_l^{(i)}\}_{l=1}^L \right) \quad \text{where} \ \ z \sim p(z), \ \ i \sim \pi_i.$$

Here, $\pi_i$ stands for the probability of selecting the $i$-th bias. This probability could be also learned using the method proposed in Khayatkhoei et al. (2018), but it is beyond our scope and we fix it as $1/A$. We denote the distribution due to this process as $p_G$.

**Encoding (Deriving Quotient Manifolds)** Due to the bias regularizer, we do not need to concretize the encoder $h_E$ by actually inverting $f_G^{(i)}$ during training. But, when encoding is needed, we can obtain the latent codes and biases by minimizing the Euclidean distance (as suggested in Ma et al. (2018)) along with a similar bias regularization as follows.

$$z, \{a_l\}_{l=1}^L = \arg \min_{\tilde{z}, \{\tilde{a}_l\}_{l=1}^L} \left\| x - f_G \left( \tilde{z}; \{\tilde{a}_l\}_{l=1}^L \right) \right\|^2 + \mu \sum_{l=1}^L \log \left\| U_l^\top a_l - U_l^\top \bar{a}_{l,\|} \right\|^2, \quad (3)$$

**Q-WGAN** Applying QMM to the Wasserstein GAN (WGAN; (Arjovsky et al., 2017)), we can define the QMM-applied WGAN losses by simply adding the regularizer:

$$\mathcal{L}_G = -E_{x \sim p_G} [D(x)] + \lambda \sum_{l=1}^L \log \left( \text{trace}(\text{cov}(U_l^\top a_l^{(i)})) \right),$$

where $\mathcal{L}_G$ and $\mathcal{L}_D = E_{x \sim p_G} [D(x)] - E_{x \sim p_R} [D(x)]$ ($p_R$ being the real data distribution) are the generator and the discriminator losses, respectively, $\lambda$ is a regularization weight, and $D(x)$ is a $k$-Lipschitz discriminator function.

**Q-$\beta$-VAE** We use a multi-generator version of $\beta$-VAE, which use EM algorithm for training (detailed in Appendix C). Applying QMM can be similarly done by adding the BA-regularizer:

$$\mathcal{L} = -E_{z \sim q(z|x)} \left[ \sum_{i=1}^A \gamma(i) \log p_G^{(i)}(x|z) \right] + \beta D_{KL} \left( q(z|x) \| p(z) \right) + R_{BA}(\{a_l^{(i)}\}_{l,i})$$

## 6 EXPERIMENTS

**Datasets** We experiment on *MNIST* (Lecun et al., 1998), *3D-Chair* (Aubry et al., 2014) and *UT-Zap50k* (Yu and Grauman, 2014) image datasets. 3D-Chair contains 1393 distinct chairs rendered from 62 different viewing angles (total 86,366 images); in experiments, only front-looking 44,576 images are used and rescaled to 64x64 grayscale images. UT-Zap50k contains images of 4 different types of shoes (total 50,025 images); the images are rescaled to 32x32.

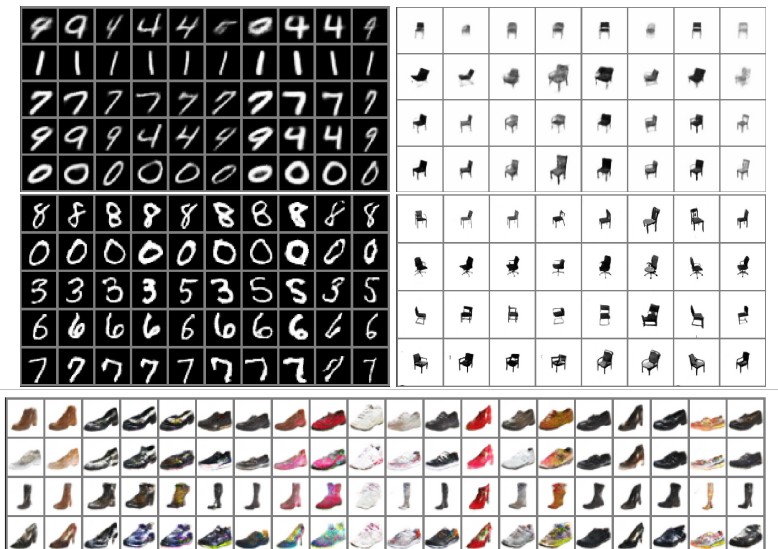

Figure 3: Images generated from the trained Q-VAEs (first row) and Q-GANs. Each $i$-th row presents samples from the $i$-th manifold (only 8 out of 20 are shown for 3D-Chair) and each column presents the samples generated from the same latent code $z$, which is randomly sampled from $p(z)$.

**Model Architectures** For GAN models, we use DCGAN (Radford et al., 2015)-like model architectures for all the datasets (see Appendix B for the complete information). In the discriminator, we use spectral normalization (Miyato et al., 2018) to achieve the $k$-Lipschitz condition. For VAE models, the architectures are generally the same as GANs, except that the batch-norms are not used. We use $\beta = 4$ and ten EM steps. In both models, Adam (Kingma and Ba, 2014) is used for training and encoding, with the default values except for the learning rate, 0.0002.

In the QMM-applied models, the number of biases, $A$, is set to 10 (MNIST), 20 (3D-Chair), and 4 (UT-Zap50k), respectively. Although our multi-biased linear layer can be applied to both fully-connected and convolutional layers, we apply it only to the former. This was sufficient for our purpose since disconnections rarely exist for such small-sized (thus low-dimensional) kernel patches.

## 6.1 Basic Multi-Manifold Learning

QMM-applied models show great performance in multi-manifold learning with a good alignment, both qualitatively and quantitatively. Looking at Fig. 3 row-wise, we can see that Q-WGAN and Q-$\beta$-VAE learn distinct manifolds well, where each manifold accounts for a different discrete class: e.g., different digits; rolling vs. armchairs; boots vs. flat shoes. Column-wise, we can see that the continuous features are well aligned among the manifolds: e.g., stroke weight, the slant of numbers; viewing angle of chairs; colors of shoes. For quantitative evaluation, we compare FID score (Heusel et al., 2017), a widely used metric to examine the diversity and quality of the generated image samples, reflecting the manifold learning performance overall. As seen in Table 1, our models give better or comparable scores than others.

## 6.2 Disentanglement of Latent Codes

To further investigate the manifold alignment performance, we examine how much the learned latent features are disentangled. We take a few images from the dataset and manually change one of the smooth features that corresponds to a known transformation (e.g., sheer transform). Then, we encode these images to the latent codes using our model, analyze the principal direction of the change, and compute the linearity of the change as the disentanglement score (see Appendix D). Fig. 4 shows that the learned continuous features are well disentangled along the principal changing directions. Table 1 shows that our models get better scores than other models.

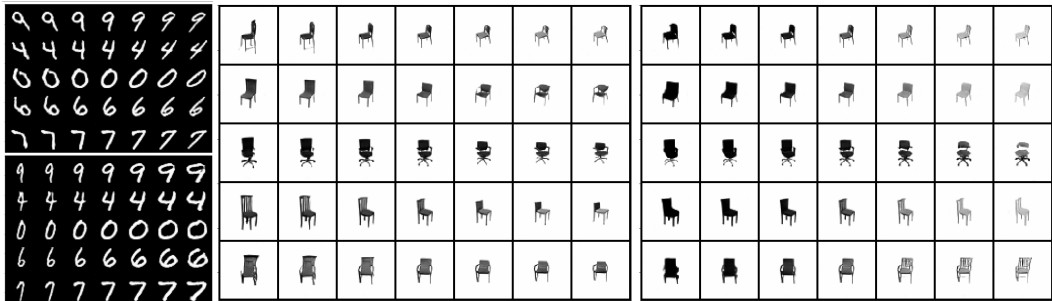

Figure 4: Disentangled features. Images are arranged the same as Fig. 3, except the columns show linear changes in the latent space along the first eigenvector from the disentanglement analysis (see Sec. 6.2). Slant, width (MNIST), height and brightness (3D-Chair) components are shown.

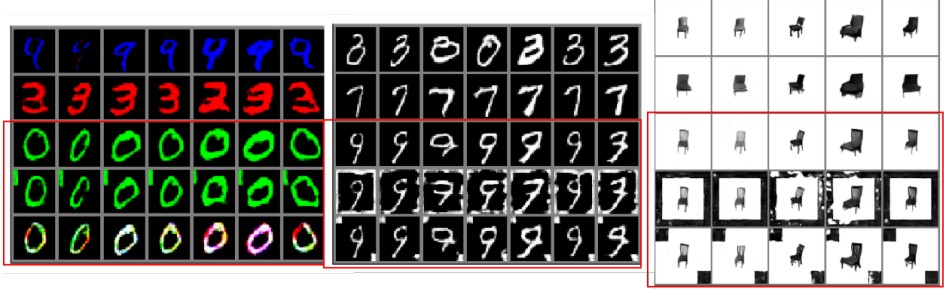

Figure 5: Manifolds derived for a noise-added data [1](highlighted in red boxes). Here, we also train on RGB-MNIST dataset: a simple extension of MNIST by coloring with (R, G, or B).

### 6.3 GENERALIZATION OVER UNTRAINED MANIFOLDS

Using the encoding method deriving quotient manifolds (Eq. 3), we examine if QMM-applied models well generalize over untrained manifolds. In both case of added noises (Fig. 5) and different datasets (Fig. 6) they indeed show fair performances in generalization, sharing generic structure with the trained manifolds (aligning continuous features column-wise).

## 7 CONCLUSION

We proposed QMM that performs multi-manifold learning in consideration for the generic structure. Unique to QMM is that it utilizes the generalization ability of a deep encoder, from which it showed its potential to infer the untrained manifolds even across different datasets. If it is trained with larger datasets such as ImageNet in the future, we expect QMM-applied models would become a more versatile tool that can derive the manifold structure of images in wild.

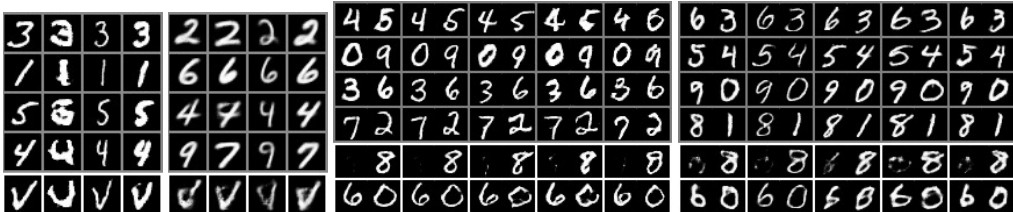

Figure 6: *Left*: QMM-applied models trained on MNIST generates alphabet 'v' included in EM-NIST dataset. *Right*: QMM-applied models trained on two-digit MNIST (multiples of 9) dataset, generating one-digit MNIST and non-multiples.

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

## A    The Proof of Theorem 1

**Theorem 1.** *The following inequality holds*

$$trace\left(cov(U^\top a^{(i)})\right) \geq \frac{1}{d_z} H(\{\lambda_k\}_{k=1}^{d_z}) trace\left(cov(a_\parallel^{(i)})\right)$$

*where $\{\lambda_k\}_{k=1}^{d_z}$ are the eigenvalues of $U^\top U$ and $H(\cdot)$ denotes a harmonic mean.*

*Proof.* Note that

$$\begin{aligned}
\text{trace}\left(\text{cov}(a_\parallel^{(i)})\right) &= \text{trace}\left(\text{cov}(U(U^\top U)^{-1}U^\top a^{(i)})\right) \\
&= \text{trace}\left(\frac{1}{A-1}\sum_{i=1}^{A} U(U^\top U)^{-1}U^\top (a^{(i)} - \bar{a})(a^{(i)} - \bar{a})^\top U(U^\top U)^{-1}U^\top\right) \\
&= \text{trace}\left(\frac{1}{A-1}\sum_{i=1}^{A} U^\top (a^{(i)} - \bar{a})(a^{(i)} - \bar{a})^\top U(U^\top U)^{-1}\right) \\
&= \text{trace}\left(\text{cov}(U^\top a^{(i)})(U^\top U)^{-1}\right) \\
&\leq \text{trace}\left(\text{cov}(U^\top a^{(i)})\right)\text{trace}\left((U^\top U)^{-1}\right),
\end{aligned}$$

where the second and the fourth lines use the definition of the covariance, third line is obtained from the cyclic property of trace and the last line is obtained from the Cauchy-Schwarz inequality of the positive semi-definite matrices. Thus,

$$\begin{aligned}
\text{trace}\left(\text{cov}(U^\top a^{(i)})\right) &\geq \text{trace}\left(\text{cov}(a_\parallel^{(i)})\right) / \text{trace}\left((U^\top U)^{-1}\right) \\
&= \frac{1}{D} H(\{\lambda_d\}_{d=1}^{D})\text{trace}\left(\text{Var}(a_\parallel^{(i)})\right)
\end{aligned}$$

where $\{\lambda_d\}_{d=1}^{D}$ are the eigenvalues of $U^\top U$ and $H(\cdot)$ denotes the harmonic mean.    □

## B    Model Architecture and Experimenting Environments

We used machines with one NVIDIA Titan Xp for the training and the inference of all the models.

### B.1    MNIST

We use $A = 10$ distinct decoding biases in the model. In the training, we set the regularization weight $\lambda = 0.05$ and use the Adam optimizer with learning rate 0.0002. In the encoding, we use the Adam optimizer with learning rate 0.1, and the set the regularization weight $\mu = 0.1$.

Table B.1: Q-WGAN architecture used for MNIST dataset (Q-$\beta$-VAE architecture is similar. See below for the differences.)

| Generator | Discriminator |
|---|---|
| Input(8) | Input(1,28,28) |
| Full(1024), BN, LReLU(0.2) | Conv(c=64, k=4, s=2, p=1), BN, LReLU(0.2) |
| Full(6272), BN, LReLU(0.2) | Conv(c=128, k=4, s=2, p=1), BN, LReLU(0.2) |
| ReshapeTo(128,7,7) | ReshapeTo(6272) |
| ConvTrs(c=64, k=4, s=2, p=1), BN, LReLU(0.2) | Full(1024), BN, LReLU(0.2) |
| ConvTrs(c=32, k=4, s=2, p=1), BN, LReLU(0.2) | Full(1) |
| ConvTrs(c=1, k=3, s=1, p=1), Tanh | |

### B.1.1 Notes on the Other Compared Models

Overall, we match the architecture of other models with our model for fair comparison. Some differences to note are:

- **DMWGAN**: We used 10 generators. Each generator has the same architecture as ours except the number of features or the channels are divided by 4, to match the number of trainable parameters. Note that 4 is the suggested number from the original paper.

- **InfoGAN**: Latent dimensions consist of 1 discrete variable (10 categories), 2 continuous variables and 8 noise variables.

- $\beta$-**VAE & Q-**$\beta$-**VAE**: We used the same architecture as the generators of Q-WGAN, except the BatchNorm layers are removed. We used the Bernoulli likelihood.

### B.2 3D-Chair

We use $A = 20$ distinct decoding biases in the model. In the training, we set the regularization weight $\lambda = 0.05$ and use the Adam optimizer with learning rate 0.0002. In the encoding, we use the Adam optimizer with learning rate 0.1, and the set the regularization weight $\mu = 0.1$.

Table B.2: Q-WGAN architecture used for 3D-Chair dataset.

| Generator | Discriminator |
|---|---|
| Input(10) | Input(1,64,64) |
| Full(256), BN, LReLU(0.2) | Conv(c=64, k=4, s=2, p=1), BN, LReLU(0.2) |
| Full(8192), BN, LReLU(0.2) | Conv(c=128, k=4, s=2, p=1), BN, LReLU(0.2) |
| ReshapeTo(128,8,8) | Conv(c=128, k=4, s=2, p=1), BN, LReLU(0.2) |
| ConvTrs(c=64, k=4, s=2, p=1), BN, LReLU(0.2) | ReshapeTo(8192) |
| ConvTrs(c=32, k=4, s=2, p=1), BN, LReLU(0.2) | Full(1024), BN, LReLU(0.2) |
| ConvTrs(c=16, k=4, s=2, p=1), BN, LReLU(0.2) | Full(1) |
| ConvTrs(c=1, k=3, s=1, p=1), Tanh | |

### B.2.1 Notes on the Other Compared Models

- **DMWGAN**: We used 20 generators. Each generator has the same architecture as ours except the BatchNorms are removed and the number of features or the channels are divided by 4, to match the number of trainable parameters. Note that 4 is the suggested number from the original paper. Note that this was the best setting among what we have tried (division number 2; ones with the BatchNorms).

- **InfoGAN**: Latent dimensions consist of 3 discrete variables (20 categories), 1 continuous variable and 10 noise variables.

- $\beta$-**VAE & Q-**$\beta$-**VAE**: We used the same architecture as the generators of Q-WGAN, except the BatchNorm layers are removed. We used the Bernoulli likelihood.

### B.3 UT-Zap50k

We use $A = 4$ distinct decoding biases in the model. For the regularization weight in the training, we start with $\lambda = 5e - 6$ then raise to $\lambda = 5e - 4$ after 300 epochs.

Table B.3: Q-WGAN architecture used for UT-Zap50k dataset.

| Generator | Discriminator |
|---|---|
| Input(8) | Input(3,32,32) |
| Full(512), BN, LReLU(0.2) | Conv(c=128, k=4, s=2, p=1), BN, LReLU(0.2) |
| Full(1024), BN, LReLU(0.2) | Conv(c=256, k=4, s=2, p=1), BN, LReLU(0.2) |
| Full(8192), BN, LReLU(0.2) | Conv(c=512, k=4, s=2, p=1), BN, LReLU(0.2) |
| ReshapeTo(512,4,4) | ReshapeTo(8192) |
| ConvTrs(c=256, k=4, s=2, p=1), BN, LReLU(0.2) | Full(1024), BN, LReLU(0.2) |
| ConvTrs(c=128, k=4, s=2, p=1), BN, LReLU(0.2) | Full(512), BN, LReLU(0.2) |
| ConvTrs(c=64, k=4, s=2, p=1), BN, LReLU(0.2) | Full(1) |
| ConvTrs(c=3, k=3, s=1, p=1), Tanh | |

## C   Q-$\beta$-VAE MODEL

Q-$\beta$-VAE model adopts a multi-generator (multi-decoder) version of $\beta$-VAE along with the BA-regularizer. To maximize the marginal likelihood given the multiple generators, it uses an EM-like algorithm:

**E-Step:**

$$Q(G) = \sum_{i=1}^{A} \gamma(i) \log p_G^{(i)}(x|z), \quad \text{where} \quad \gamma(i) = p_G^{(i)}(x|z)/\sum_i p_G^{(i)}(x|z)$$

**M-Step:**

$$\mathcal{L} = -E_{z\sim q(z|x)}\left[Q(G)\right] + \beta D_{KL}\left(q(z|x)\|p(z)\right) + R_{BA}(\{a_l^{(i)}\}_{l,i}).$$

where $R_{BA}(\{a_l^{(i)}\}_{l,i}) = \lambda \sum_{l=1}^{L} \log\left(\text{trace}(\text{cov}(U_l^\top a_l^{(i)}))\right)$.

In the E-Step, it takes the expectation over the different generators; the responsibilities of each generator can be computed as presented above since $\gamma(i) = p(i|x,z) = \pi_i p_G^{(i)}(x|z)/\sum_i \pi_i p_G^{(i)}(x|z) = p_G^{(i)}(x|z)/\sum_i p_G^{(i)}(x|z)$. In the M-Step, we plug in the computed expectation $Q(G)$ as the marginal likelihood term ($\gamma(i)$ is fixed). Repeating E and M step multiple times (we take ten repeats), we finish the gradient step for a single mini-batch.

## D   DISENTANGLEMENT SCORE

To compute the disentanglement score, we first take 500 images from the dataset and manually change one of the smooth features that corresponds to a known transformation. For example, we change the slant of the MNIST digits by taking a sheer transform. With 11 different degrees of the transformation, we obtain 5500 transformed images in total. We encode these images to obtain the corresponding latent codes and subtract the mean for each group of the images (originates from the same image) to align all the latent codes. Then, we conduct Principal Component Analysis (PCA) to obtain the principal direction and the spectrum of variations of the latent codes. If the latent features are well disentangled, the dimensionality of the variation should be close to one. To quantify how much it is close to one, we compute the ratio of the first eigenvalue to the second eigenvalue of the PCA covariance, and set it as the disentanglement score.

## E    EFFECT OF THE NUMBER OF GENERATORS, $A$

To investigate the effect of the number of generators (or biases), $A$, we train our model on MNIST with different $A$ values then compute the FID and the disentanglement scores (Fig. E.1). It can be seen that our model performs consistently better than the baseline, WGAN, regardless of the different $A$ values.

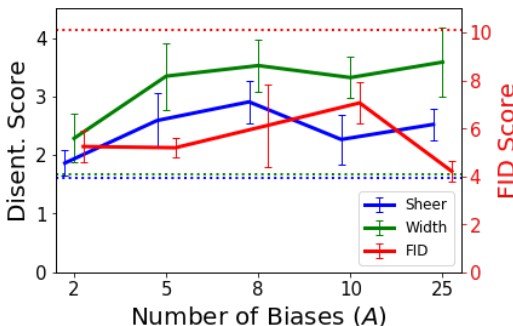

Figure E.1: The FID scores (right axis, the smaller the better) and the disentanglement scores (left axis, the larger the better) of Q-WGAN with varying $A$ are shown for MNIST dataset. The dashed lines show the mean scores of the baseline model (WGAN).

## F    EFFECT OF THE REGULARIZATION WEIGHT, $\lambda$, IN TRAINING

To investigate the effect of the regularization weight, $\lambda$, we train our model on MNIST with different $\lambda$ values. It can be seen that our model performs consistently better than the other compared models—Q-WGAN with no regularization ($\lambda = 0$), DMWGAN, MADGAN-like (see the next paragraph)—regardless of the different $\lambda$ values (other models are omitted for better readability; see Table 1 for the omitted ones). It can be also seen that the scores are not very sensitive to the different choices of $\lambda$'s; this is beneficial in that one may choose any reasonable value for $\lambda$ when training the model with a new dataset.

Here, *MADGAN-like* is a DMWGAN model, but has a similar parametrization to the MADGAN [6]: The parameters of the first three layers (from the top) are shared for all the generators. In contrast, Q-WGAN shares the parameters in all the layers except the biases in the fully-connected layers. Thus, in a sense, one can say that Q-WGAN shares only the last few layers, whereas *MADGAN-like* shares only the first few layers (of course, there is another difference due to the regularizers). Although the both models have the shared structures among the generators, Q-WGAN performs much better in both of the scores as seen in Figure F.2. This suggests that a simple parameter sharing is not enough to obtain a good performance, and the bias regularizer is indeed required.

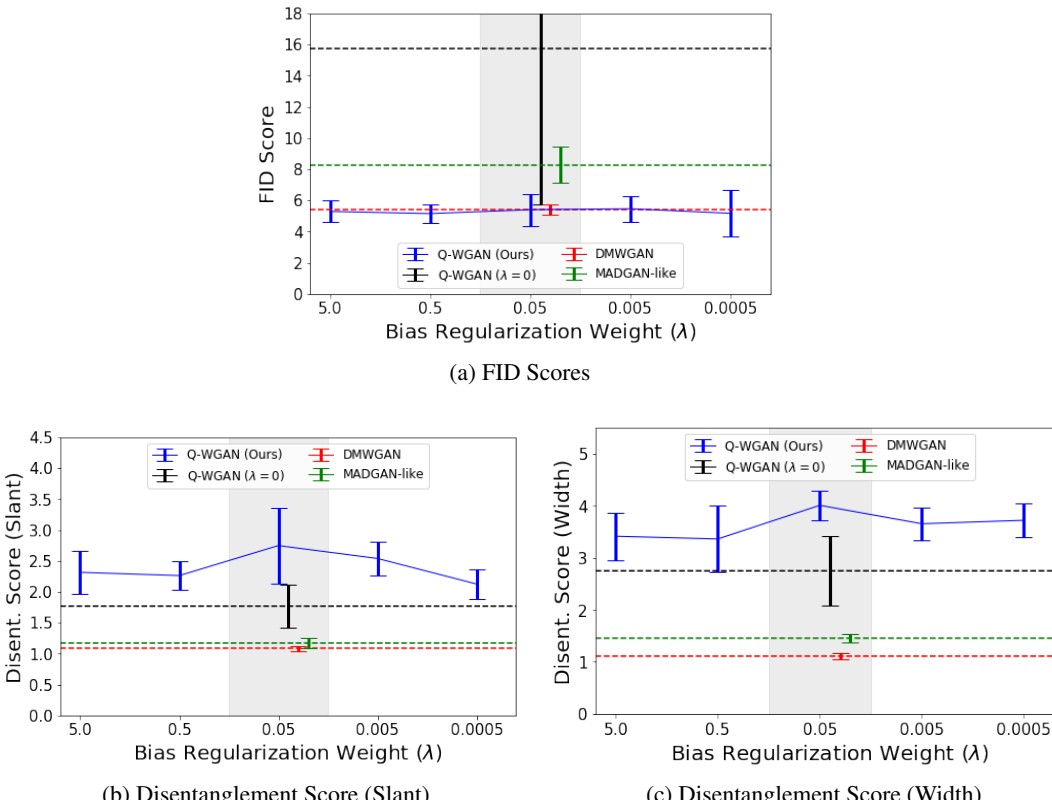

(a) FID Scores

(b) Disentanglement Score (Slant)

(c) Disentanglement Score (Width)

Figure F.2: The FID scores (the smaller the better) and the disentanglement scores (the larger the better) of Q-WGAN with varying $\lambda$ are shown for MNIST dataset. Note the other models are positioned in the center (shaded in gray) to be visually comparable with the best-performing Q-WGAN model ($\lambda = 0.05$).

## G  EFFECT OF THE REGULARIZATION WEIGHT, $\mu$, IN ENCODING

To investigate the effect of the regularization weight, $\mu$, in encoding (Eq. 3), we take a trained model and encode an image from the train set using different $\mu$ values. Then, plugging in the encoded (estimated) biases, $\{a_l\}_{l=1}^L$, we randomly generate the samples from this new manifold and compare the qualities for different $\mu$'s.

From Figure G.3 (b), we can see that the quality of the encoding tends to improve as $\mu$ gets smaller. This might seem opposite to what we expect, as weaker regularization gives better results. However, looking closer, we can see that the features like the slant and the stroke are more aligned with a stronger regularization of $\mu = 0.05$, when comparing with the other manifolds in the bottom pane. Thus, a trade-off exists here between the image quality of the samples and the alignment of the manifold to the others. Note we chose to use $\mu = 0.05$ in all the other experiments.

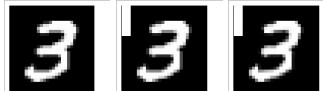

(a) **Left**: An image taken from the training dataset. **Middle**: The same image with a rectangle noise. **Right**: Regenerated image from the estimated biases, $\{a_l\}_{l=1}^L$, and the estimated latent code $z$, from Eq. 3.

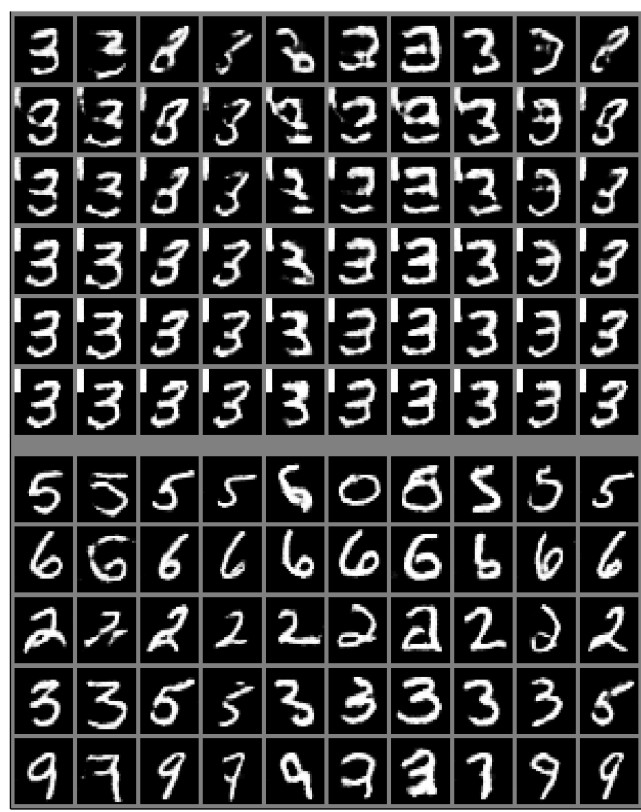

(b) **Top pane**: Generated samples from the estimated biases, $\{a_l\}_{l=1}^L$. The biases are estimated with different $\mu$'s, (5.0, 0.5, 0.05, 0.005, 0.0005, 0.0), from the top to the bottom. **Bottom pane**: Generated samples from the originally-learned biases, $\{\{a_l^{(i)}\}_{i=1}^A\}_{l=1}^L$ (only 5 out of 10 are shown). Note the images in the same column have the same latent value $z$.

Figure G.3: Effect of the regularization weight $\mu$ in encoding is shown with a MNIST-trained Q-WGAN model.

## H    EFFECT OF THE BIAS REGULARIZER

To examine the effectiveness of our bias regularizer, we visualize the raw values of biases $\{a_l^{(i)}\}_{i,l}$ and their (pseudo-)tangential component $\{U_l^\top a_l^{(i)}\}_{i,l}$ (see Fig. H.4, H.5). In all figures, we see that the biases are diverse, but their tangential components are well aligned due to the bias regularizer (left). On the contrary, without the regularizer, the tangential components are not aligned (right).

## I    SAMPLES GENERATED FROM VARIOUS MODELS

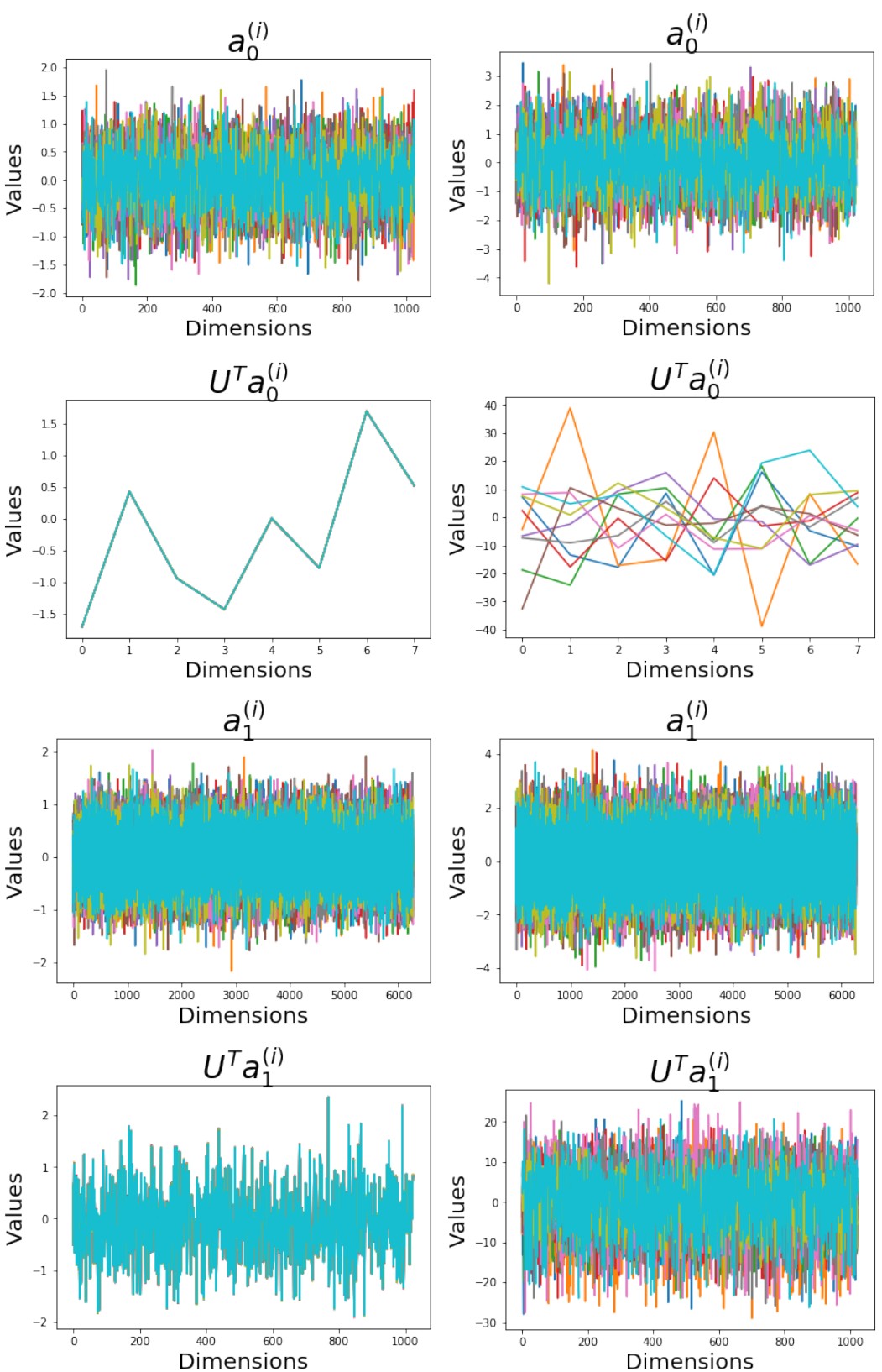

Figure H.4: Biases $a_l^{(i)}$ and their (pseudo-)tangential components $U_l^\top a_l^{(i)}$ of the Q-WGAN models, trained on ***MNIST***. Individual curve indicates each $i$-th bias. **Left** Parameters of Q-WGAN **Right** Parameters of Q-WGAN without the regularizer ($\lambda = 0$). It can be seen that the regularizer makes the tangential components of the biases well aligned.

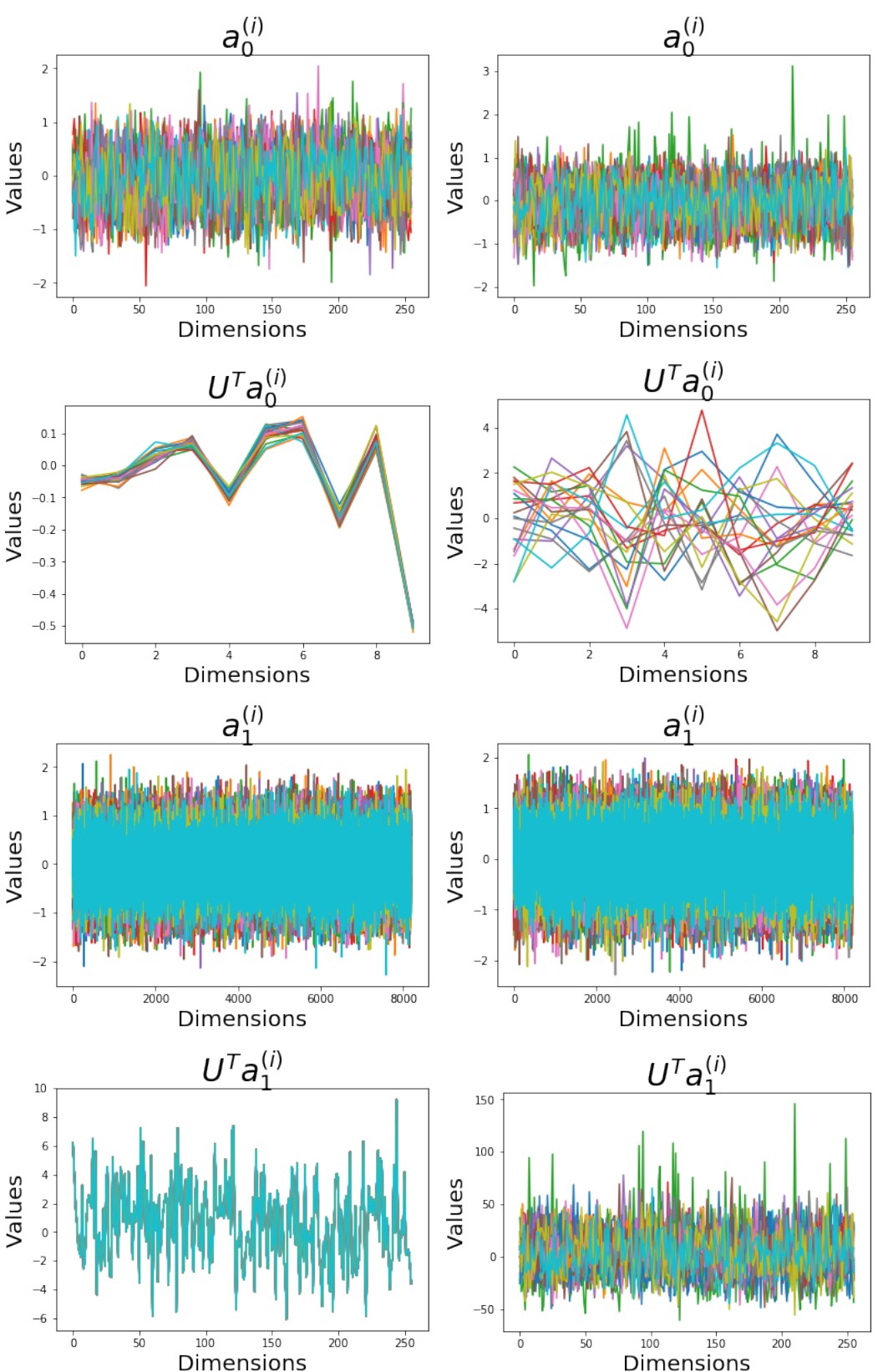

Figure H.5: Biases $a_l^{(i)}$ and their (pseudo-)tangential component $U_l^\top a_l^{(i)}$ of the Q-WGAN models, trained on **3D-Chair**. Individual curve indicates each $i$-th bias. **Left** Parameters of Q-WGAN **Right** Parameters of Q-WGAN without the regularizer ($\lambda = 0$). It can be seen that the regularizer makes the tangential components of the biases well aligned.

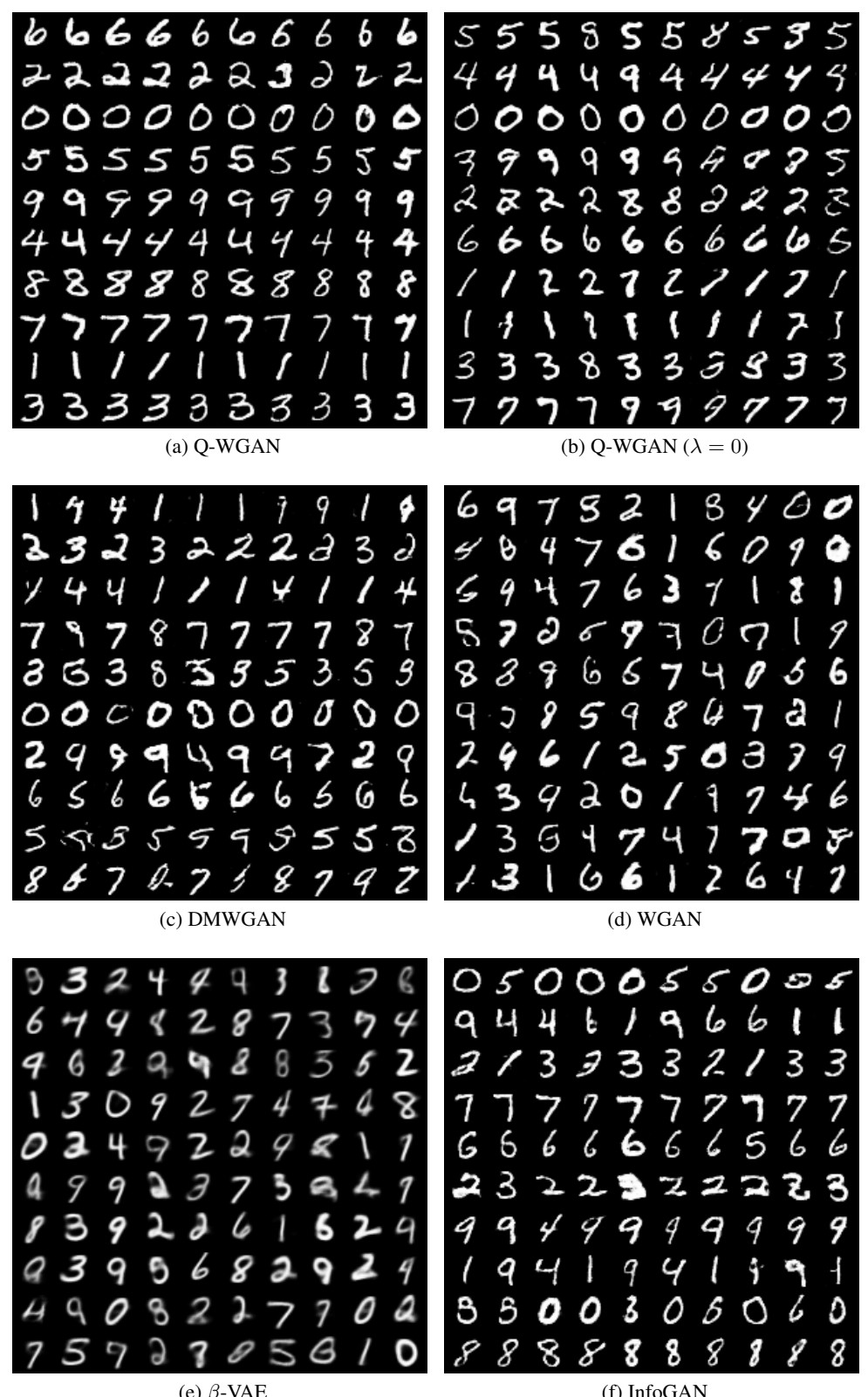

(a) Q-WGAN

(b) Q-WGAN ($\lambda = 0$)

(c) DMWGAN

(d) WGAN

(e) $\beta$-VAE

(f) InfoGAN

Figure I.6: MNIST image samples generated from the trained models

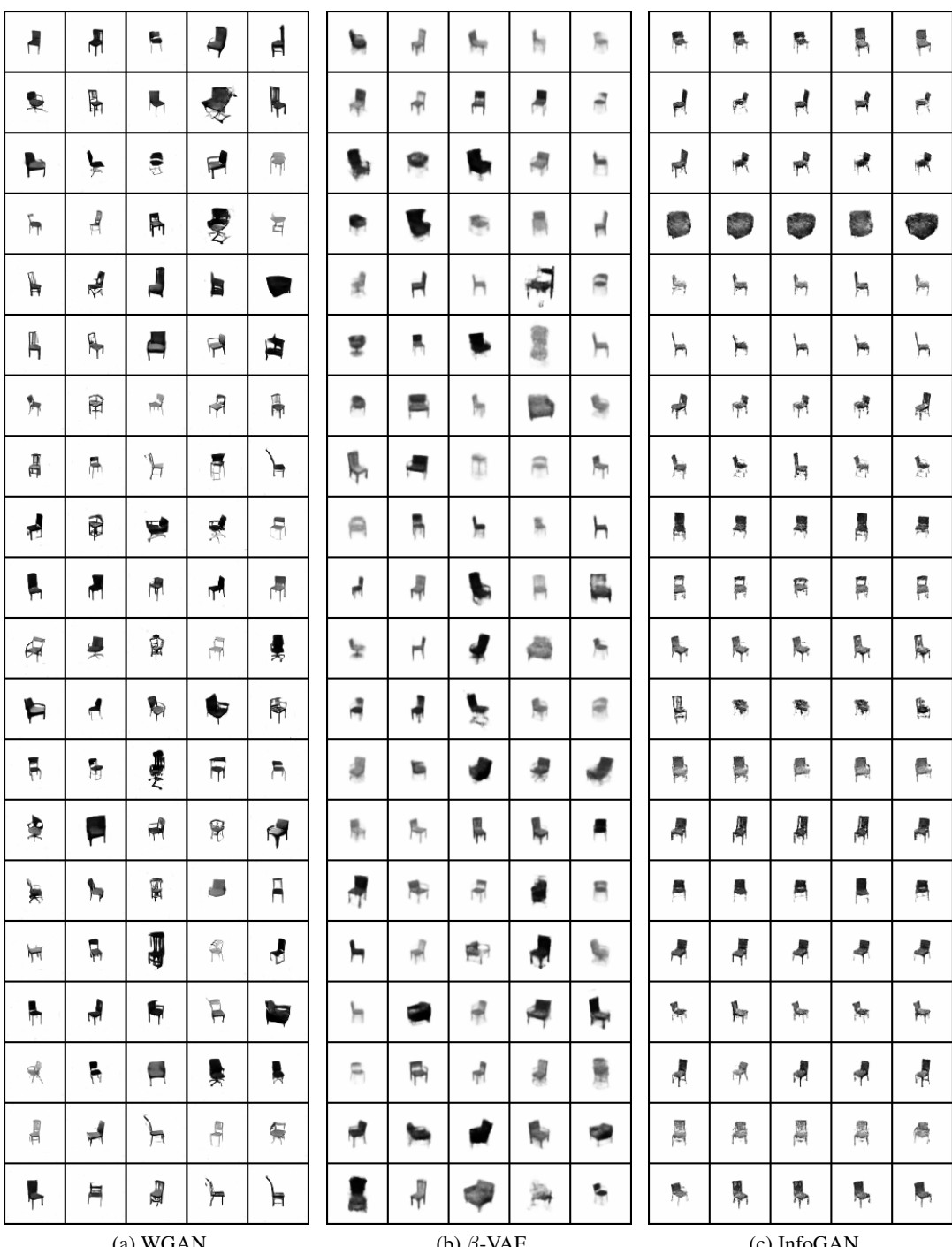

(a) WGAN      (b) $\beta$-VAE      (c) InfoGAN

Figure I.7: 3D-Chair image samples generated from the trained models

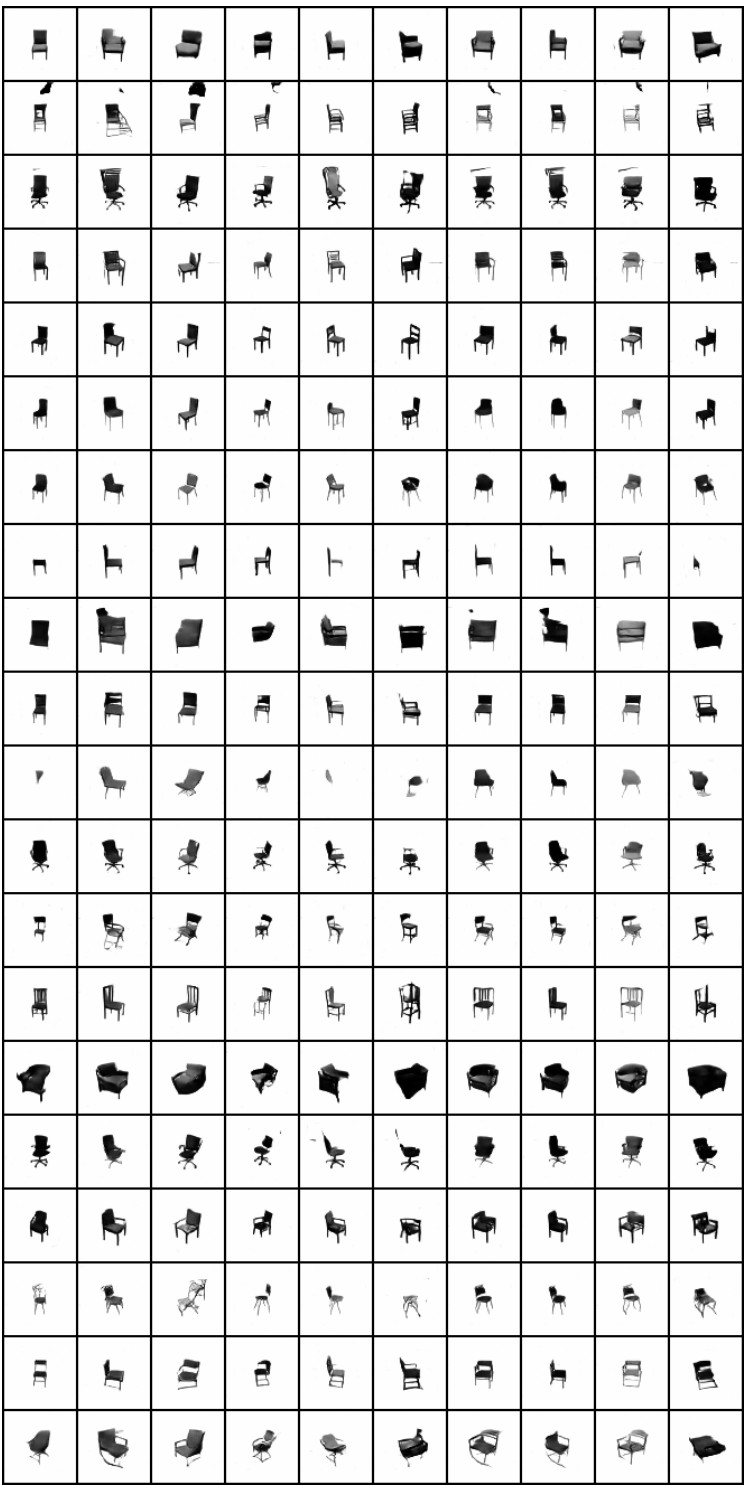

Figure I.8: 3D-Chair image samples generated from the trained Q-WGAN

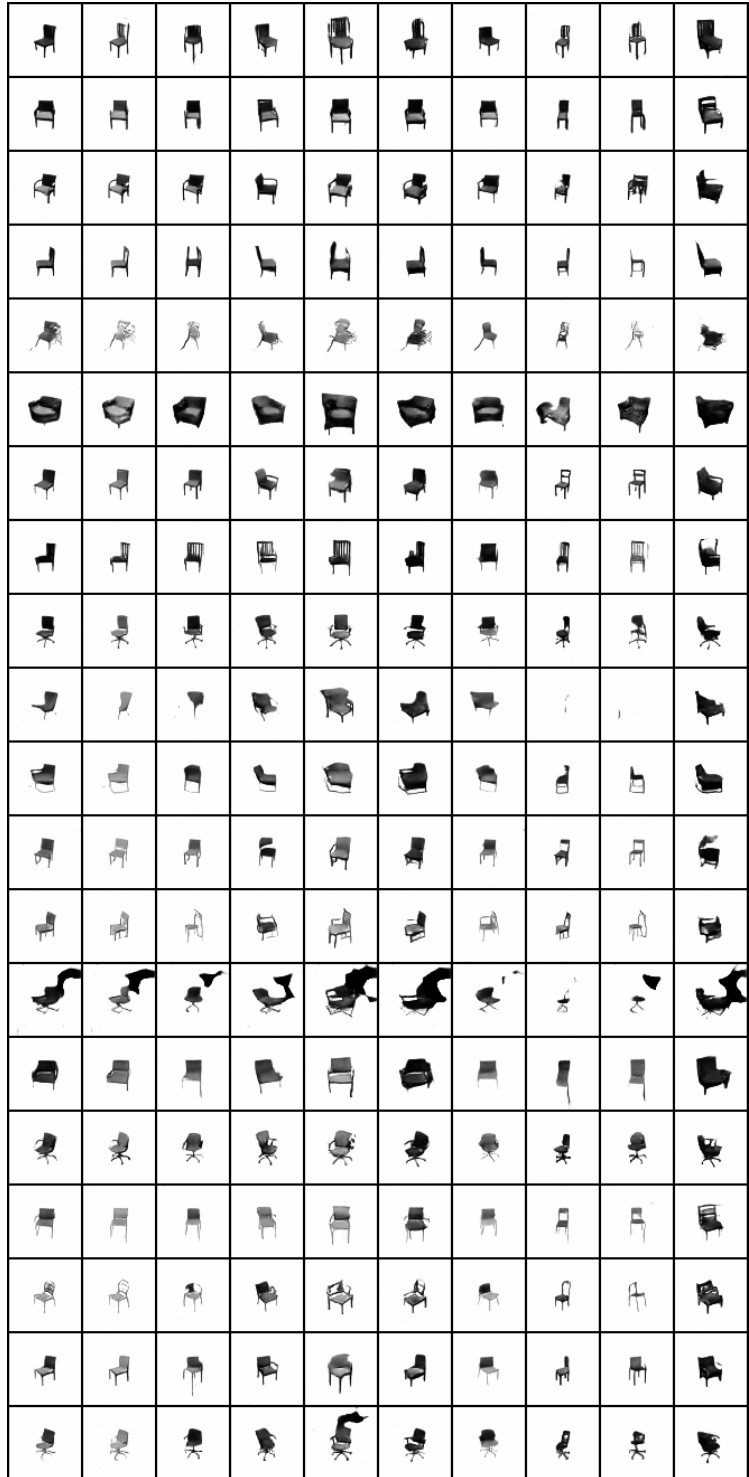

Figure I.9: 3D-Chair image samples generated from Q-WGAN ($\lambda = 0$)

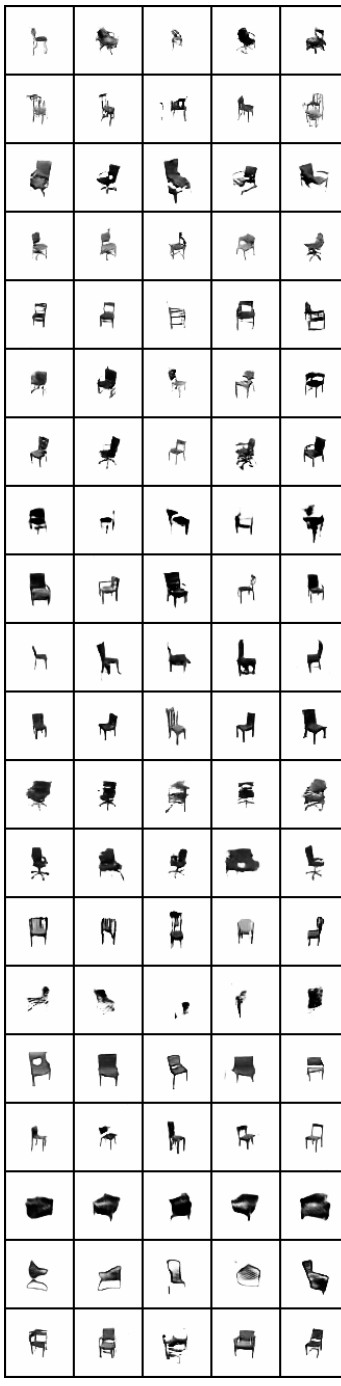

Figure I.10: 3D-Chair image samples generated from the trained DMWGAN

