# OpenReview forum: "Deep Quotient Manifold Modeling"
_ICLR.cc/2021/Conference — Reject_

### Official Review · AnonReviewer1 · 2020-10-28
**Official Blind Review #1**

**Rating:** 4
**Confidence:** 5

**Review:**

_Summary_:
The author extends generative models with multi-generators by restricting the generators to share weights and all bias to be regularized in order to enforce that the inverse maps of the generators can be represented by a single encoder. The regularizer proposed minimizes an upper bound the the sum of the bias variances. This extension is evaluated on a set of visual datasets (MNIST, 3DChair and UT-Zap50k) with respect to density estimation (evaluated with the FID score) and disentanglement (evaluated with their own disentanglement score).

_Strengths_:
- The method proposes a regularizer which is easy to understand and implement. This makes it easy to apply to existing generative models.
- The evaluation showed that sample diversity w.r.t. FID score is comparable/superior to baseline generative models.

_Weaknesses_:
For me, the main problems are the motivation of encoder compatibility, the clarity of the paper and limited evaluation:
1) Motivation of encoder compatibility: I don't understand why the inverse of all generators needs to represented by one (!) single encoder? Why can't the generators be in general invertible and thus there exists a mapping for each observed variable to the latent space? What is the advantage of one encoder?
2) The paper needs to improve its preciseness and clarity. I found it at parts confusing to understand because it introduces terms without really explaining or justifying them, e.g., "encoder compatibility", "good generalizability", "equivalence relation". Further, background is lacking to give details about the model that this paper extends on. This also hinders readability of the section where the authors introduced their own method. Further, assumption of continuity and generalizability have been made without defining these terms or reference.
3) The evaluation is limited to generators w.r.t. generator architecture (fully-connected layers only). Further, FID score show high standard deviation and thus are not necessarily superior compared to baselines and the introduced disentangle metric are only reported for specific attributes of MNIST and 3D-Chair.

More detailed comments are below.

_Overall assessment_:
Unfortunately, I do not believe this paper is ready for acceptance. Therefore I am recommending a rejection. I do believe it is an interesting approach. And with improvement in terms of clarity and evaluation, I am happy to improve my score if the authors can make the necessary improvement.

_Detailed comments and questions_:
- "since discrete features are both common and combinatorial" (Introduction): Common to what?
- "they usually have the same generic structure since the underlying continuous features since the underlying continuous features remain the same" (Introduction): Has this been shown in any work before (if so, please cite)? I can also imagine this being depending on the specific learning task.
- "it induces a generalizable equivalence relation between data, and the manifold structure of out-of-sample data can be derived by taking the quotient of this relation" (Introduction): What is a generalizable equivalence relation between data? Is the quotient of this relation defined somewhere?
- "Since deep encoders usually exhibit good generalizability" (Introduction): What is the generalizability considered with respect to? Can you elaborate?
- Multi-generator scheme (Sections 2.2, 3): Can you elaborate on the actual model used for Quotient Manifold Modelling (QMM)? The paper mentions that a multi-generator scheme is used but there are no more details than that. It would be helpful to know what the learning objective is, how the (multiple) generator (and discriminator) are used.
- "Let H be a set of encoding maps $(X \to Z)$ that can be represented by a deep encoder" (Section 3.1): Why is being represented by one encoder important?
- "this binding is meaningful only when H has a certain property": Which one? Is that the generalizability that is mentioned in the next paragraph?
- "its elements--deep encoders--have good generalizability": Similar to the claim above, what does that mean to have generalizability? How can generalizability be quantified?
- High standard deviation of FID scores of proposed methods: The proposed method have high standard deviation. Can you explain why this is? Is this an optimization problem?
- Proposed disentanglement metric: There is a vast literature of evaluation for disentanglement, why did you choose to propose your own disentanglement metric? How does it compare to the existing disentanglement metrics (advantages, similarity, etc.)?
- Table 1 (disentanglement): Can you also report overall disentanglement of the dataset? For disentanglement specifically you can also use existing benchmarks and datasets from Locatello et al. (2018).

_Minor_:
- Definition of generative maps ${f_G^{(i)}:Z->M [...]\}_i^A$ (Section 3.1): I believe $A$ was not defined before, is that just the number of generators?

_Post-Rebuttal_:
Unfortunately, the authors neither did update their paper nor addresses my comments. Therefore, I'm keeping my recommendation of rejection.

---

### Official Review · AnonReviewer2 · 2020-10-29
**Nice idea but too-simplistic approach**

**Rating:** 6
**Confidence:** 5

**Review:**

1. The "orthogonal directions  to  the  equivalence-relation  contours" in section 3.2. should be properly defined. I can understand that the authors want to say that the horizontal spaces along the sections are perpendicular to the vertical spaces along the fibers. But still need clarification for a general audience.

2. The tangential components are ``parallel'' to U and hence in prop. 1, they need to be same. This is not obvious and need to be properly defined. Along this direction, the tangent and normal of a are with respect U. On the other hand, along the fibers there exists a group operation, which is translation for a linear layer and justify Prop. 1. So Prop. 1 is not surprising or worth proving in the paper.

3. The same nonlinear activation functions (we useLeakyReLU) and batch-normalization layers constraint the explanability of the network. This implies one needs more linear layers. To me, this needs further investigation, and while reading the paper, I thought some interesting ideas will come up during developing non-linearity. Although the design using quotient structure is interesting, the choice of sharing same vertical component is a easy consequence of principal fiber bundle theory. It will be interesting to study ``better'' choice of non-linearity. Can authors comment on that?

4. The experimental setup is a bit under-developed. The disentnglement scores have only been shown on MNIST images, what about for more complicated images like CelebA, e.g., "Learning Disentangled Representations via Independent Subspaces", CVPR 2019.

5. Similar kind of simplistic setup has been used for showing generalizability on untrained datasets.

6. These kind of simplistic experimental setup points to the simplicity of just changing liner layer and using same non-linearity. Can authors please verify whether the method ``works'' (in such a simplistic manner) on more complicated datasets, i.e., do we need a very deep network? If yes is the training still stabilized or does converge?

---

### Official Review · AnonReviewer4 · 2020-10-30
**Deep Quotient Manifold Modeling**

**Rating:** 5
**Confidence:** 2

**Review:**

The authors propose a method to train deep generative models on quotient manifolds and show improved performance on some simple standard test sets.

The article makes a strong and compelling case for the need of working with quotient manifolds, and shows good results over baselines in all the test cases investigated. The method is simple enough to implement and it’s theoretical motivation plausible (although far from conclusively proven).

However, I feel the descriptions of the quotient manifolds investigated (e.g. the continuous-discrete differences) are somewhat lacking. Figure 1 strongly implies the quotient spaces are w.r.t. some continuous quantity, while the method is discrete. It does feel like the method is more aimed towards manifolds with multiple disjoint components than true quotient manifolds. The experiments are also lacking. The baselines are old, and for some of them I know for sure that there are better results available out there. I would be much happier if the ablations were one-to-one, e.g. using the exact same architecture except for this change.

Overall I find this an OK contribution, but it would be strengthened by either giving more sharp theoretical results or by using more up to date experimental methods.

Some smaller comments:
* It’s not quite clear that the regulariser will force the parallel components to behave as expected. Wouldn’t another possible way for the network to solve this be to make the U’s very small?
* The architecture seems to assume a standard feed-forward setup. This isn’t really state of the art (almost everyone does resnets). Would it be possible to extend the setup to this slightly more general setting?
* The domain of “i” should be defined
* The definition of the pseudo inverse uses only holds for full rank matrices. How do you make sure this is the case in practice?
* GANs, VAEs, etc have reasonably strong theoretical foundations and simply adding a regulariser might break them in weird ways. Would it be possible to incorporate the proposed methods more rigorously?

---

### Official Review · AnonReviewer3 · 2020-11-02
**A good paper with clear intuition and promising performance.**

**Rating:** 8
**Confidence:** 4

**Review:**

Natural images may lie on the disjoint multi manifolds rather than one globally connected manifold, and this can cause several difficulties for the training of Generative Adversarial Networks (GANs) and variational autoencoders (VAEs).
The paper proposes QMM, a new generative modeling scheme that inherits the multi-generator scheme but involves an essential regularizer enforcing the encoder compatibility. QMM considers generic manifold structure by the generalizable equivalence relation between data, thereby taking the quotient of this relation and driving the manifold structure of untrained data.

Clarity: The paper is well written and easy to follow.

Novelty:  Inheriting multiple generators scheme for generate model training has been proposed in many previous works. There are some recent work for applying multiple generators to VAEs or GANs, e.g., Pan et al., Latent Dirichlet allocation based generative adversarial networks, Neural Networks, 2020,  etc.   A major difference in this paper is that the additional regularizer enforcing the encoder compatibility and the quotient of the plausible equivalence relation.

---

### Decision · Program_Chairs · 2021-01-07
**Final Decision**

**Decision:**

Reject

**Comment:**

This paper presents an interesting method dubbed quotient manifold modeling to handle the "multi-manifold" structure of natural data and generalize to new manifolds that arise from novel discrete combinations. While some of the methods and ideas were appreciated by reviewers, there were a number of experimental and clarity concerns. The authors's did not submit a rebuttal, and the many unaddressed concerns (especially around experimental baselines) lead me to recommend rejecting this work.